# Caffeine-Containing Energy Shots Cause Acute Impaired Glucoregulation in Adolescents

**DOI:** 10.3390/nu12123850

**Published:** 2020-12-16

**Authors:** Jane Shearer, Raylene A. Reimer, Dustin S. Hittel, Mackenzie A. Gault, Hans J. Vogel, Matthias S. Klein

**Affiliations:** 1Department of Biochemistry and Molecular Biology, Faculty of Kinesiology, University of Calgary, Calgary, AB T2N 4N1, Canada; reimer@ucalgary.ca (R.A.R.); dhittel@gmail.com (D.S.H.); magault@ucalgary.ca (M.A.G.); 2Alberta Children’s Hospital Research Institute, Alberta Children’s Hospital, Calgary, AB T2N 4N1, Canada; 3Discovery DNA Inc., Calgary, AB T2N 4N1, Canada; 4Department of Pediatrics, Alberta Children’s Hospital, Calgary, AB T2N 4N1, Canada; 5Department of Biological Sciences, Faculty of Science, University of Calgary, Calgary, AB T2N 1N4, Canada; vogel@ucalgary.ca; 6College of Food, Agricultural and Environmental Sciences, The Ohio State University, Columbus, OH 43210, USA

**Keywords:** pediatrics, adolescent, adverse reaction, energy drink, metabolomics

## Abstract

Caffeine-containing, nutritionally fortified energy shots are consumed at high rates by adolescents, yet little is known about their metabolic impact. The purpose of this study was to examine the consequences of small format, caffeinated energy shots on glucose metabolism and gastrointestinal hormone secretion in adolescents. Twenty participants aged 13–19 years participated in a double-blind, randomized cross-over study consisting of two trials separated by 1–4 weeks. Participants consumed a volume-matched caffeinated energy shot (CAF, 5 mg/kg) or a decaffeinated energy shot (DECAF) followed by a 2 h oral glucose tolerance test. Blood samples were collected and area under the curve (AUC) calculated for glucose, insulin and gut and metabolic hormones. Consumption of CAF resulted in a 25% increase in glucose and a 26% increase in insulin area under the curve (AUC, *p* = 0.037; *p* < 0.0001) compared to DECAF. No impact on gut hormones was observed. To further characterize responses, individuals were classified as either slow or fast caffeine metabolizers based on an allele score. Glucose intolerance was greater in genetically fast vs. slow caffeine metabolizers and differences between groups were supported by distinct serum metabolomics separation. Consumption of caffeine-containing energy shots results in acute impaired glucoregulation in healthy adolescents as characterized by hyperinsulinemia following an oral glucose challenge.

## 1. Introduction

Caffeine-containing energy drinks and small format energy shots (CED) are marketed to reduce fatigue and increase energy and alertness [1]. Along with caffeine, these beverages typically contain large amounts of other active ingredients, including vitamins, amino acids, flavorings and antioxidant compounds. An estimated 74% of youth consume energy drinks with 16% consuming more than two a day [2]. It is well recognized that energy drinks may be especially harmful for children, whose lower body mass may expose them to caffeine concentrations above recommended limits. Children often experience adverse impacts of energy drinks to a greater extent than adults for this reason [3]. CED consumption is associated with childhood obesity [4,5], likely impairs insulin sensitivity [6] and causes sleep disturbances in adolescents [7,8].

From a metabolic perspective, it is accepted that co-administration of a carbohydrate load in the presence of caffeine impairs whole body glucose disposal by 20–30% in adults [9]. Reductions in whole body glucose disposal have been shown during oral glucose tolerance tests (OGTT) and euglycemic-hyperinsulinemic clamps [10,11,12]. Caffeine-induced glucose intolerance occurs in a dose-dependent manner at concentrations as low as 1 mg caffeine/kg body weight, with no apparent threshold [13]. Habitual consumption of caffeine does not appear to result in an adaptation to this response [14]. As the half-life of caffeine is in the range of 4–6 h, insulin action could be impaired for some hours after ingestion. Given these data, one serving of a typical energy drink or small format energy shot would be expected to not only exceed daily recommended limits for caffeine for children (Health Canada, 2.5 mg/kg), but also impair insulin sensitivity for most of their waking hours. However, it must be acknowledged that there are genetic contributors to caffeine metabolism, and that individual responses may vary [15]. To examine the impact of CED on glucose regulation, adolescents consumed either a caffeine-containing energy shot (5 mg/kg) or its equivalent decaffeinated version 40 min prior to an oral glucose tolerance test. Additional assessments included gut and metabolic hormones known to influence glucoregulation as well as a preliminary analysis of genetic contributors to the observed responses.

## 2. Materials and Methods

### 2.1. Participants

This study was approved by the Conjoint Health Research Ethics Board at the University of Calgary (REB14-1093) and registered at ClinicalTrials.gov (NCT03512496). Subjects were recruited by convenience sampling through local advertising. Twenty (10 males, 10 females) adolescents aged 13–19 years without underlying medical conditions, supplement use, medication intake affecting glucose tolerance, oral contraceptive use and without phenylketonuria or known caffeine allergies were invited to participate.

### 2.2. Test Beverages

The composition of the decaffeinated beverage (5-h Energy Decaffeinated, DECAF) and the caffeine-containing beverage (5-h Energy Original, Regular Strength, CAF) are shown in Table 1. Both are sugar-free and administered as pre-determined mixtures in identical cups and of equivalent volumes, color and taste (both were sucralose-sweetened). Each subject consumed 5 mg/kg of caffeine, corresponding to one 57 mL (commercially available) serving for a typical 13-year-old (~38 kg).

### 2.3. Experimental Design

Following parental consent and child assent, basic demographic information was collected and questionnaires were self-administered to collect information regarding physical activity levels, caffeine consumption habits, and medical history. Anthropometric measures including height, weight, waist circumference and body-fat determination via dual-energy X-ray absorptiometry were collected. Participants completed trials in a randomized and double-blind fashion, each trial separated by 1–4 weeks. Although diet was not strictly controlled, participants were instructed to record and consume the same meal prior to each trial. Participants also arrived at the laboratory following a 24 h abstention from caffeine and vigorous exercise and after an overnight fast (10–12 h) to undergo an oral glucose tolerance test (OGTT). Upon arrival, participants were given a brief questionnaire to screen for conditions that may exclude them from the blood draw. After confirming compliance, an indwelling catheter was inserted into an antecubital vein and a fasting blood sample was collected. Immediately after, the treatment drinks: DECAF or CAF were consumed. Trial order was randomly assigned by a computer-generated randomizing program. However, we did not control for menstrual cycle phase as previous work has shown that menstrual cycle has no effect on the absorption, distribution, metabolism and elimination of caffeine [16]. Participants, research assistants and the research nurse performing the OGTT were blinded throughout the trial. Following treatment, subjects waited for 40 min before a second blood sample was taken (time: 0 min), immediately followed by the ingestion of a glucose drink (1.75 g/kg body weight Trutol, to a maximum of 75 g). Additional blood samples were drawn at 30, 45, 60, 90 and 120 min. Data on participant characteristics and the DECAF trial have been previously reported [17] and are included here to provide context to the novel data on CAF administration. In this study, characteristics of males and females are reported separately.

### 2.4. Blood Analysis

At each time point, blood was collected in a vacutainer treated with EDTA (BD Vacutainer^®^) and placed on ice immediately for later glucose analysis via the glucose oxidase method using a colorimetric assay (Cayman Chemical Company, Ann Arbor, MI, USA). In addition, EDTA tubes were treated with diprotinin-A (0.034 g/L blood; MP Biomedicals), sigma protease inhibitor (1 g/L blood; Sigma-Aldrich, St. Louis, MO, USA) and Roche Pefabloc (1 g/L of blood) for analysis of gut and metabolic hormones. After blood collection, samples were centrifuged and the supernatant was collected and frozen at −80 °C for later determination. Metabolic-related hormones were simultaneously quantified by a multiplex assay (Millipore, St. Charles, MO, USA) according to the manufacturer’s protocol. Analytes included C-Peptide, ghrelin, glucose-dependent insulinotropic peptide (GIP), glucagon-like peptide-1 (GLP-1) (active), glucagon, insulin, leptin and pancreatic polypeptide YY (PYY) (total). The assay sensitivities of these markers ranged from 0.6–87 pg/mL.

### 2.5. Combined Caffeine Sensitivity Allele Score

All participants ejected 5 mL of saliva into a tube for genetic analysis (Oragene, DNA Genotek, Ottawa, ON, Canada). Immediately following collection, tubes were sealed and stored until shipment. Analyses were run on data collected before December 2016. DNA extraction and genotyping were performed on saliva samples by an accredited laboratory (Laboratory Corporation of America). Comprehensive genotyping (933 202 SNP Chip, Illumina OmniExpress Plus Genotyping Bead Chip, 23andMe, Sunnyvale, CA, USA) was conducted as per manufacturer standard [18]. Five analytically validated variants involved in caffeine metabolism (rs4410790, rs2470893, rs2472297, rs2472299 and rs762551) were selected for analysis based on previous methods [19].

### 2.6. Serum Metabolomics Analysis

To gain insight into metabolic differences arising from either slow or fast caffeine metabolism, metabolomics analysis was performed on 39 serum samples collected at 120 min of each OGTT. A single sample was not able to be assessed due to volume restrictions. Samples were analyzed as previously described by ^1^H-NMR spectroscopy [20]. Briefly, serum samples were filtered in Amicon 10 kDa cutoff filters (Millipore, Billerica, MA, USA). Four hundred microliters of filtered serum was mixed with 200 µL phosphate buffer and 50 µL deuterated water and measured by means of 1D ^1^H-NMR spectroscopy on an Avance II 600 MHz spectrometer (Bruker BioSpin, Milton, ON, Canada). Data were analyzed in R 3.5.1 (R Foundation for Statistical Computing, Vienna, Austria). Spectra were split into bins of 0.01 ppm width in the region between 9.5 and 0.5 ppm, and bin size was manually increased for signals that visibly changed positions between spectra. After removal of water and noise areas, 472 signals were left for analysis. Signal intensities were scaled by the probabilistic quotient normalization (PQN) method. General linear models (GLM) were calculated individually for each signal, using caffeine (DECAF/CAF) and caffeine metabolism score (slow/fast) to model the observed metabolite concentration. Adding an interaction term (caffeine: caffeineMetabolism) did not reveal any metabolites significantly changing effect direction through such interactions; therefore, this term was not used for further analysis. Resulting *p*-values were corrected using false discovery rate controlling at an FDR level of 20%. Metabolites were identified using database queries (https://hmdb.ca/, https://bmrb.io/) and measurements of pure standard compounds.

### 2.7. Statistical Analysis

All other analyses were done using GraphPad Prism 7 (San Diego, CA, USA). Descriptive statistics are presented as mean and standard deviation (SD) for numerical/continuous variables and percentages for categorical variables. Area under the curve (AUC) for serial measurements of glucose and gut and metabolic hormones during the two hour OGTTs (time: 0–120 min) was calculated using the trapezoidal method [21]. Repeated measures ANOVA for parametric data and Friedman’s test for non-parametric data were employed. If data violated the assumptions of sphericity, a Greenhouse-Geisser correction was utilized. Where significant treatment effects were found, a Sidak multiple hypothesis test identified which treatments were significantly affected. Paired *t*-tests were applied to explore if the insulin sensitivity index [22], insulin and glucose AUC differed between DECAF and CAF treatments. To examine the genetic influence on glucose response, an allele score was calculated for each participant as previously described by Nordestgaard et al. [19]. For each genotype, scores of 0, 1 and 2 were assigned and further binned into either low (slow caffeine metabolism) (1–5) or high (fast caffeine metabolism) (6–10) categories. A two-way ANOVA and Sidak multiple hypothesis test was used to examine differences between these groups. Where applicable, residuals of the robust fit were analyzed to identify outliers. This method uses an outlier test adapted from the false discovery rate approach of testing for multiple comparisons using the ROUT method of regression set at Q = 1% [23]. *p* < 0.05 was considered significant.

## 3. Results

### 3.1. Participant Characteristics

A total of twenty adolescents (10 male, 10 female) participated in this study with a self-reported Tanner stage of 4.3 ± 0.8 (mean ± SD) (Table 2). No significant effects of sex, age or BMI on glucose or insulin AUC were found (*p* > 0.05); therefore, data from all participants were pooled. All fasting blood glucose and insulin levels were within normal ranges and there were no differences in baseline samples between treatments (*p* > 0.05). All participants reported exercising regularly with 40% of participants exercising 2–4 days a week and the remaining 60% exercising 5–7 days a week. The majority of participants (*n* = 17, 85%) reported that they consumed caffeine.

### 3.2. Impact of Caffeine-Containing Energy Shots on Glucose and Insulin Responses

Glucose and insulin OGTT curves are shown in Figure 1. Compared to DECAF, significant increases in glucose concentrations were observed with CAF at 30, 45, 60 and 120 min during the OGTT (*p* < 0.05, Figure 1A). Examination of mean AUC showed consumption of CAF resulted in a 25% increase in glucose excursion with values of 556.9 ± 26.8 and 683.8 ± 31.4 (mmol/L*120 min^−1^) for DECAF and CAF, respectively (*p* < 0.0001) (Figure 1A). When individual responses were compared between DECAF and CAF treatments (AUC), the majority of participants (18/20) showed an exaggerated glucose response when caffeine was consumed (Figure 1B). Insulin levels were not significantly different at any time point examined (*p* > 0.05) (Figure 1C). However, the mean AUC was greater following CAF treatment with a 26% increase in insulin excursion with values of 42,437.2 ± 4711.1 and 52,324.5 ± 7371.2 pmol/L*120 min^−1^ for DECAF and CAF, respectively (*p* = 0.037) (Figure 1D). Analysis of individual change between DECAF and CAF revealed larger inter-individual variation with 15/20 participants showing an increase in overall insulin responses based on AUC (Figure 1D) with CAF. The insulin sensitivity index (ISI) [22] was also affected by treatment (5.66 ± 0.48, 4.62 ± 0.45, for DECAF and CAF, *p* = 0.0016) and was significantly lower following CAF treatment, indicating insulin resistance due to CAF consumption (Figure 1E). Presence or absence of caffeine was confirmed by a serum sample obtained at 120 min. Results showed a concentration of 0.31 ± 0.5 μmol/L and 22.9 ± 1.5 μmol/L for DECAF and CAF trials, respectively (*p* < 0.001) (Figure 1F).

### 3.3. Impacts of Caffeine-Containing Energy Shots on Gut and Metabolic Hormone Secretion

A summary of all hormone results are presented in Table 3. No differences were noted between CAF and DECAF treatments for glucagon, leptin, PYY, ghrelin, GIP or GLP-1 (*p* > 0.05).

### 3.4. Combined Caffeine Sensitivity Allele Score

Due to the limited and underpowered sample size of this study, genetic variants of interest were analyzed as a composite allele score [19]. Briefly, five genetic variants combined into an allele score yielding scores between 0 and 10. Scores were then binned into either low (slow caffeine metabolism, 1–5) or high (fast caffeine metabolism, 6–10) for analysis. In total, 9 participants were classified as having a low allele score, while the remaining 11 subjects were classified as having a high score. Details of the analysis and variants present in participants are shown in Table 4. No differences in baseline glucose or insulin levels were found between slow and fast allele scores prior to the CAF and DECAF trials (*p* > 0.05, data not shown). Results of this analysis showed individuals with a high allele score experienced greater glucose intolerance as shown by significant differences in glucose and insulin excursion as well as a trend in the ISI with *p* = 0.07 (Figure 2). In contrast, only glucose excursion was different between CAF and DECAF treatments in those individuals with a low allele score. This difference appears to be associated with insulin secretion and peak insulin levels that were greater in high vs. low allele score individuals (*p* < 0.05, data not shown).

### 3.5. Serum Metabolomics

General linear model (GLM) analysis revealed strong and significant metabolic impacts of caffeine consumption. This is visualized in Figure 3A,B by principal component analysis (PCA) of significant metabolites, showing clear separation between DECAF vs CAF treatments as well as between slow and fast caffeine metabolizers. A PCA analysis of the full dataset can be found in Appendix A. Specifically, serum caffeine, 1,7-dimethylxanthine, folic acid and lactic acid were positively correlated to caffeine intake, while serum tartaric acid and choline decreased with caffeine intake (Figure 3C,D, Appendix A).

Consistent with our metabolic results, serum glucose levels were elevated in fast vs. slow metabolizers with CAF. Levels of circulating caffeine as well as its primary metabolite 1,7-dimethylxanthine (paraxanthine) were found to be significantly higher in individuals with fast metabolism scores. Other distinct metabolites included citric acid that was higher in fast metabolizers, while proline levels were decreased in this group.

## 4. Discussion

Caffeine-containing energy shots represent a new and relatively understudied area of nutrition [24]. In the present study we evaluated the metabolic impact of these small format beverages in adolescents, who represent key consumers of the products [25]. Very little is known about the metabolic effects of caffeine energy shot consumption in adolescents who by nature of the normal influences of puberty are generally more insulin resistant [26]. Our results show CAF consumption, equivalent to one sugar-free small format energy shot containing ~5 mg/kg caffeine, induces insulin resistance in adolescents compared to a control, DECAF beverage. Specifically, caffeine resulted in a ~25% increase in glucose and insulin excursion. Insulin resistance resulting from caffeine consumption was also evident in examination of the ISI [22] that calculates whole body insulin sensitivity, through OGTT-derived glucose and insulin levels.

Our results are in line with other studies in adults showing that co-administration of alkaloid caffeine [10,11,27,28] or caffeinated coffee [29,30,31,32] in combination with a carbohydrate load causes acute insulin resistance. However, the response to caffeine in coffee is attenuated compared to caffeine alone, presumably due to the presence of polyphenols and other coffee-related constituents [33,34]. This response is dose-dependent [13] and occurs independently of previous caffeine exposure [14]. Analysis of the impact of caffeine on glucose tolerance shows that on average, caffeine impairs glucose tolerance (evaluated by either an oral glucose tolerance test or insulin clamp) in the range of 30% [6]. Numerous theories as to the mechanism by which caffeine impairs glucose tolerance at physiological concentrations have been proposed. Among these, the antagonism of adenosine receptors by caffeine (a potent, non-specific adenosine receptor antagonist) predominates, with evidence showing that caffeine directly impacts both insulin and exercise-stimulated glucose uptake in skeletal muscle [35,36,37].

In the present study, the energy shots were commercially available preparations in CAF and DECAF formats. The ingredient profile of the two beverages was similar, mainly differing in their caffeine concentration. This study design allowed the impact of caffeine in energy shots to be determined, but the impact of other energy shot ingredients should not be overlooked. Basrai and colleagues [38] conducted an elegant study wherein the impact of a large volume (1000 mL), caffeine-containing energy drink was studied on both cardiovascular and metabolic responses in young adults. Results demonstrated adverse impacts of the energy drink on blood pressure, prolongation of the corrected QT interval (QTc) interval as well as insulin sensitivity [38]. The caffeinated energy drink consisting of 32 mg/100 mL caffeine, 400 mg/100 mL taurine and 31 mg/100 mL glucuronolactone resulted in elevations in insulin that caused a greater modified homeostatic model assessment of insulin resistance (HOMA-IR) score, indicating greater insulin resistance. Of note, changes in HOMA-IR resulting from energy drink consumption in the study were greater than caffeine administration alone, as was the QTc interval prolongation [38]. This highlights a potentially synergistic impact of energy drink ingredients, an issue that has been previously raised in the energy drink literature and may lead to more adverse effects than traditional caffeinated beverages such as coffee [39,40,41].

As an individual’s preference, sensitivity and metabolism of caffeine are relatively stable over a lifetime and determined in part by genetics [42], we conducted a preliminary analysis to examine the role of genetic variation in caffeine-induced insulin resistance. Because this analysis was based on a small number of subjects, individuals were classified as having either a high or low allele score for variation of *AHR*, *CYP1A1* and *CYP1A2* genes (SNPs: rs762551, rs4410790, rs2470893, rs2472297 and rs2472299) as described by Nordestgaard [19]. Individuals with high allele scores, representative of fast caffeine metabolism, had significantly greater disruption to glucose and insulin secretion as well as impaired insulin sensitivity throughout the OGTT compared to individuals with a low allele score. A number of theories have been proposed to explain these findings. Keijzers et al. [43] theorize that tolerance to caffeine-induced insulin resistance may occur with chronic consumption, but individuals who are fast metabolizers of caffeine may be unable to develop this tolerance due to the short caffeine half-life. In agreement, our metabolomics results show both caffeine and its metabolite 1,7-dimethylxanthine (paraxanthine) were present in greater levels compared to slow metabolizers with CAF treatment at 120 min of the OGTT.

Likewise, it is also possible that caffeine metabolites contribute to its impact on glucose intolerance. Caffeine is rapidly and primarily metabolized by liver CYP1A2 and CYP2E1 into its three related dimethylxanthines; 1,3-dimethylxanthine (theophylline, 11%), 3,7-dimethylxanthine (theobromine, 4%) and 1,7-dimethylxanthine (paraxanthine, 80%), through a series of demethylation steps [44]. These metabolites are also metabolically active and can influence glucose tolerance. Employing hyperinsulinemic-hypoglycemic clamps, De Galan et al. [45] demonstrated that theophylline infusion (2.8 mg/kg) causes profound insulin resistance as shown by a reduction in glucose infusion rates in both healthy controls and type 1 diabetes. Given this, the greater concentrations of caffeine metabolites in those with fast metabolism may explain in part why their glucose tolerance was affected to a greater extent with CAF consumption. Taken together, results of our study suggest that fast caffeine metabolizers who consume CED on a regular basis likely experience greater impairment in glucose tolerance compared to slow metabolizers. As such, it is likely that the ~30% decline in insulin sensitivity seen with caffeine consumption in adults across dozens of studies minimizes the true impact of caffeine in this subgroup. Our overall finding that caffeine-induced glucose tolerance is affected by an individual’s genetics adds to a growing body of evidence showing that caffeine preference, ergogenic potential, impacts on sleep and even disease-related susceptibility are genetically influenced [15,46].

Several strengths and limitations need to be considered in the present study. A key strength is the examination of adolescents and commercially available energy shots in relation to glucose tolerance. In this study, the beverage was administered as a mixture and was sugar-free followed by a carbohydrate load (OGTT). While caffeine and the carbohydrate load were administered separately in our study to examine glucose tolerance, the majority of energy drinks (as opposed to small volume shots) are consumed as sugar-sweetened mixtures. It must also be acknowledged that the results from one beverage may not be applicable across the entire CAF beverage category given the wide range of ingredients and caffeine concentrations that are available. It is also likely that there are interactions between ingredients. In particular, concerns regarding an interaction between caffeine and taurine found in CAF have been highlighted in previous reports [47,48].

Limitations include venous blood sampling (vs. arterial), lack of a standardized diet prior to trials as well as the administration of an OGTT that does not reflect what happens in everyday life where mixed meals are consumed. Additional limitations include the genetic analysis, which is based on a small number of subjects and a combined allele score. These results are preliminary and require confirmation at the individual genotype level. However, strong separation of slow and fast allele scores by metabolomics analyses adds strength to the applicability and validity of the approach. Next, all participants were healthy and within normal body mass ranges. Therefore, the clinical relevance of acute insulin resistance caused by caffeine in this demographic is unknown. It is very likely that responses are exaggerated in adolescents with obesity and/or insulin resistance.

## 5. Conclusions

CED are popular and frequently consumed among children and adolescents, an impressionable and understudied population. In the present study, we provide evidence that caffeine administered as part of a complex energy shot mixture causes acute, transient insulin resistance in adolescents. The clinical relevance of this finding warrants further investigation.

## Figures and Tables

**Figure 1 nutrients-12-03850-f001:**
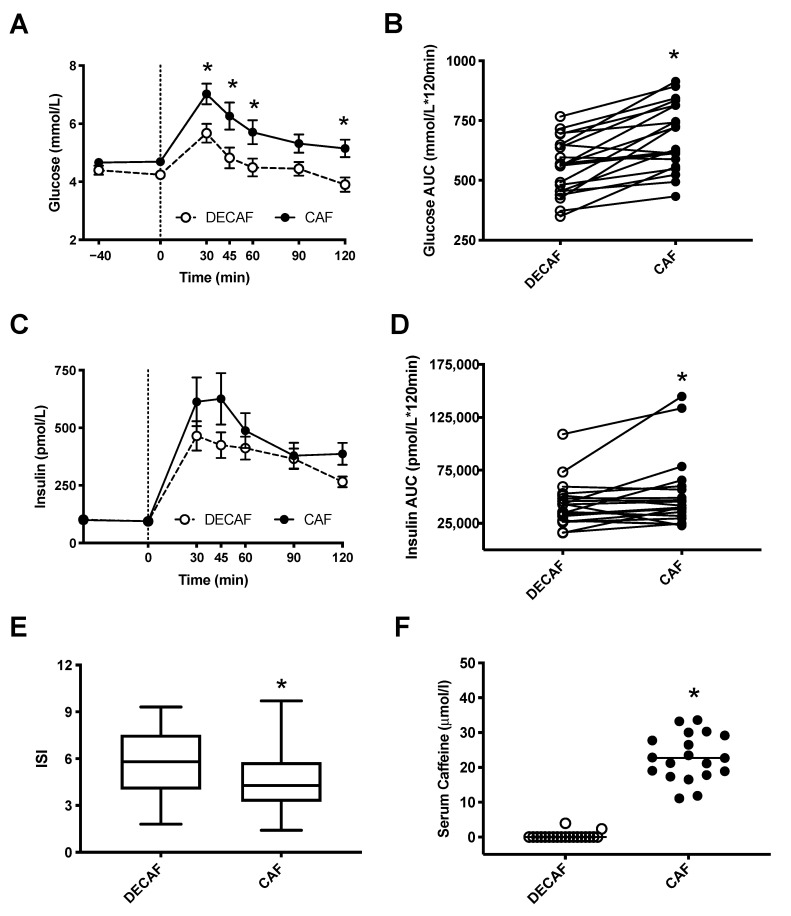
Postprandial glucose and insulin responses following ingestion of either a decaffeinated (DECAF) or caffeine-containing (CAF, 5 mg/kg) energy shot at −40 min followed by an oral glucose tolerance test (OGTT) at 0 min (indicated by a dashed line). (**A**) Blood glucose excursion curves. (**B**) Individual changes in glucose area under the curve calculated by the trapezoidal method (AUC). (**C**) Insulin excursion curve. (**D**) Individual changes in insulin AUC. (**E**) Calculated Matsuda insulin sensitivity index (ISI) [22] for each treatment (95% confidence interval). (**F**) Mean serum caffeine concentrations obtained at 120 min of each OGTT. All data are shown as means ± SE, *n* = 20 subjects in a randomized, cross-over design. * Indicates *p* < 0.05, a statistically significant difference between treatments.

**Figure 2 nutrients-12-03850-f002:**
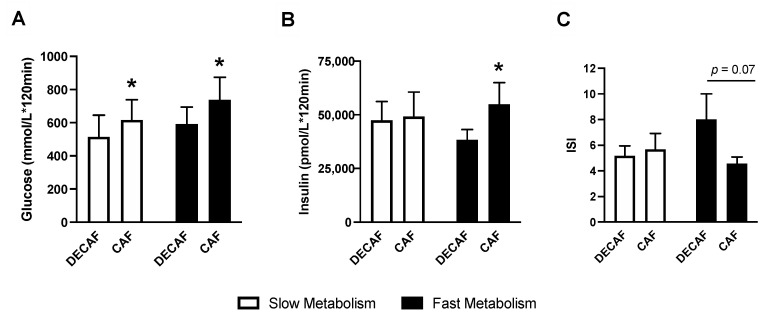
Genetic influences of caffeine-containing energy shot consumption on glucose and insulin tolerance. Subjects were classified as having either a low (slow caffeine metabolism, *n* = 9) or high (fast caffeine metabolism, *n* = 11) allele score based on variation at rs762551, rs4410790, rs2470893, rs2472297 and rs2472299 as described by Nordestgaard [19]. (**A**) Blood glucose AUC, (**B**) Insulin AUC, (**C**) Matsuda insulin sensitivity index (ISI) [22] in participants classified with slow or fast caffeine metabolism. All data are shown as means ± SE, *n* = 20 subjects in a randomized, cross-over design. * Indicates *p* < 0.05, a statistically significant difference between treatments.

**Figure 3 nutrients-12-03850-f003:**
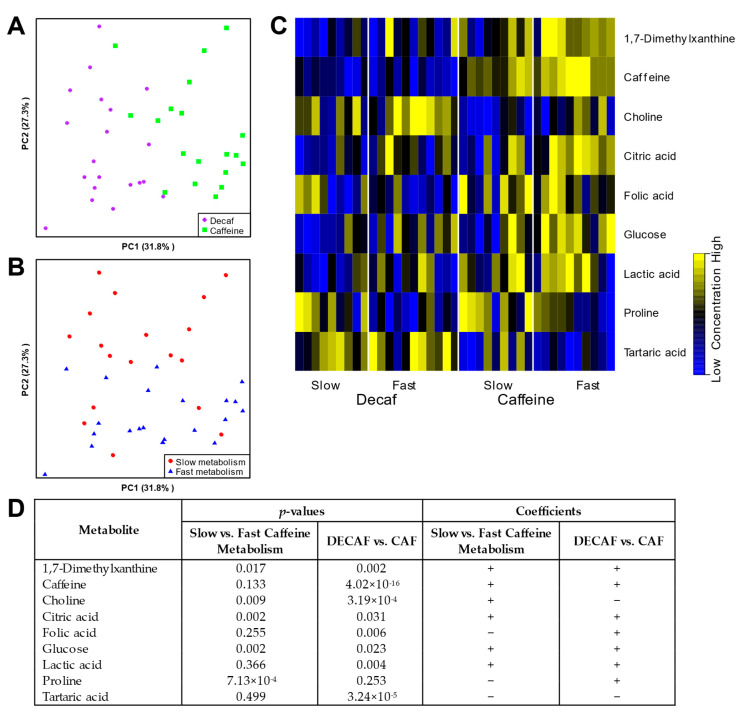
Serum metabolomics (^1^H-NMR) of 120 min time point of each OGTT. (**A**) Principal component analysis (PCA) showing separation of DECAF and CAF treatments. (**B**) PCA analysis showing separation of slow and fast allele scores. (**C**) Heatmap depicting significant changes between both DECAF and CAF treatments as well as slow and fast allele scores. (**D**) Table of serum metabolites significantly correlating with caffeine intake and/or caffeine metabolism score along with their *p*-values and direction (either + or −) of coefficient change.

**Table 1 nutrients-12-03850-t001:** Ingredient profiles of the caffeine-containing CAF (5-h Energy original) and DECAF (5-h Energy Decaffeinated) shots.

5-h Energy Original (CAF, Regular Strength)			5-h Energy Decaf (DECAF)		
Ingredient	Amount	%Daily Value	Ingredient	Amount	%Daily Value
Vitamin B6	40 mg	2000%	Vitamin B6	40 mg	2000%
Folic Acid	400 mcg	100%	Folic Acid	400 mcg	100%
Vitamin B12	500 mcg	8333%	Vitamin B12	500 mcg	8333%
Sodium	18 mg	<1%	Sodium	18 mg	<1%
Niacin	30 mg	150%	Niacin	0 mg	0%
Caffeine	215 mg	n/a	Caffeine	6 mg	n/a
Energy Blend	1870 mg	n/a	Energy Blend	2009 mg	n/a
(Taurine, glucuronic acid, malic acid, N-Acetyl L-tyrosine, L-phenylalanine, caffeine, citicoline)			(Taurine, choline, glucuronic acid, N-Acetyl L-tyrosine, L-Phenylalanine, malic acid, caffeine)		
Other Ingredients: Purified water, natural and artificial flavors, sucralose, potassium sorbate, sodium benzoate, EDTA.			Other Ingredients: Purified water, natural and artificial flavors, sucralose, potassium sorbate, sodium benzoate, EDTA.		

**Table 2 nutrients-12-03850-t002:** Participant characteristics.

	Males	Females
*n*	10	10
Age (years)	16.4 ± 2.2	17.5 ± 2.2
Height (m)	1.75 ± 0.1	1.64 ± 0.1
Weight (kg)	72.3 ± 24.6	59.3 ± 4.3
BMI	23.1 ± 6.0	22.2 ± 2.2
BP (Systolic)	133 ± 15	111 ± 8
BP (Diastolic)	71 ± 9	62 ± 9
% Fat	17.3 ± 4.7	24.2 ± 4.5
% Lean	79.4 ± 4.3	72.4 ± 4.4

Data are shown separately for males and females (mean ± SD).

**Table 3 nutrients-12-03850-t003:** Gut and metabolic hormones area under the curve (AUC) response to treatments.

Measure	Treatment	AUC Mean (SE)	*p*-Value
Ghrelin (pg/mL/120 min)	DECAF	5208.2 (519.2)	*p* = 0.945
	CAF	5173.6 (615.3)	
PYY (pg/mL/120 min)	DECAF	7530.1 (868.4)	*p* = 0.412
	CAF	7771.1 (864.2)	
Leptin (pg/mL/120 min)	DECAF	536,471.8 (72,667.4)	*p* = 0.246
	CAF	489,282.6 (67,250.3)	
GIP (pg/mL/120 min)	DECAF	2960.6 (565.8)	*p* = 0.915
	CAF	3003.4 (517.0)	
GLP-1(pmol/L/120 min	DECAF	407.0 (47.8)	*p* = 0.438
	CAF	383.2 (40.4)	
Glucagon (ng/L/120 min)	DECAF	3085.8 (508.1)	*p* = 0.998
	CAF	2963.2 (347.7)	
C-peptide (nmol/L/120 min)	DECAF	115.0 (7.2)	*p* = 0.241
	CAF	120.1 (6.7)	

Values for each trial, decaffeinated 5-h energy (DECAF) and caffeinated 5-h energy (CAF) and their corresponding *p*-values are shown. Abbreviations are as follows: Gastric inhibitory polypeptide, GIP; Glucagon-like peptide-1, GLP-1; Peptide YY, PYY.

**Table 4 nutrients-12-03850-t004:** Combined Caffeine Sensitivity Allele Score.

SNP	GENE	CHR	POSITION	GMAF	ALLELE	SCORE	COUNT
					TT	0	2
rs4410790	*AHR*	13	17284577	C > T, 0.387	CT	1	11
					CC	2	7
					CC	0	9
rs2470893	*CYP1A1-4011*	15	75019449	C > T, 0.278	TC	1	10
					TT	2	1
					CC	0	12
rs2472297	*CYP1A1-12441*	15	75027880	C > T, 0.225	TC	1	8
					TT	2	0
					AA	0	3
rs2472299	*CYP1A1*	15	74741059	G > A, 0.350	GA	1	5
					GG	2	12
					CC	0	3
rs762551	*CYP1A2-163*	15	75041917	A > C, 0.674	AC	1	5
					AA	2	12

A total of five alleles were used to calculate a combined score as previously described [19]. Low allele score (1–5) represents combination of genotypes associated with slow caffeine metabolism; high allele score represents combination of genotypes contributing to fast caffeine metabolism (6–10). Count (far right column) represents the genotypes of the participants. Abbreviations are as follows: aryl hydrocarbon receptor (AHR); chromosome (Chr); cytochromes P450 (CYP); minor allele frequency (MAF); single nucleotide polymorphism (SNP).

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
