# Peer review of "Caffeine-Containing Energy Shots Cause Acute Impaired Glucoregulation in Adolescents"

_nutrients, 2020, doi:10.3390/nu12123850_

Round 1
Reviewer 1 Report
Here Shearer et al examine the acute effects of ingestion of caffeine-containing energy drinks on glycemic control in adolescents. The findings reveal that prior consumption of caffeine-containing energy drinks exacerbates the glucose and insulin response during an OGTT. Overall, the study is clearly written and presented. I have the following comments that the authors are encouraged to consider.
- Was any diet control implemented in the day preceding each of the experimental trials? If not, this is an important limitation as it is possible that differences in pre-trial diet have contributed to observed effects attribute to CAF. This should be acknowledged in the limitations section.
- The authors should consider providing the rationale for the metabolomics studies – this is not apparent. Also, it would be beneficial to outline why the 120 min sample was analysed as opposed to a fasting sample or other time points during the OGTT.
- In addition to presenting the AUC data for the hormones in Table 2, it would be beneficial to show the time course data during the OGTT – similar for the glucose and insulin data in Figure 1.
- It is interesting that the C-peptide response (AUC data Table 2) was similar between the CAFF and DECAF trials. This indicates that insulin secretion was not affected by CAFF. Thus, despite the higher glucose levels during the OGTT in the CAFF trial, insulin secretion was not exacerbated. As one would expect the higher glucose response to cause greater insulin secretion, does this indicate that CAFF could somehow be impairing beta cell function? Have the authors considered using approaches such as the C-peptide minimal model to examine this? This point seems worthy of some discussion in the manuscript.
- The other point that these C-peptide results highlight is that the elevated insulin response caused by CAF must be mediated by effects on insulin clearance rather than insulin secretion. This point should also be explored. In this regard, it might be worth calculating indices of insulin clearance.
- Detail about the method for determining ISI is missing. Please include in the methods section.
- In relation to the limitations section, there are perhaps a couple of other points that could be considered. Firstly, venous blood was sampled, not arterialised venous blood from a heated hand vein. This could influence the glucose measurements. Secondly, an OGTT was used. While the OGTT has clinical relevance, its relevance to everyday life is somewhat limited since it is devoid of fat and protein, both of which can affect glycemic and hormone responses.
Author Response
REVIEWER 1. We wish to sincerely thank the reviewer for their efforts in evaluating this manuscript, especially during the COVID pandemic that has increased stress levels worldwide. Your comments and attention to detail have increased the clarity and quality of our manuscript. All comments have been addressed and where possible changes to the manuscript have been made.
Was any diet control implemented in the day preceding each of the experimental trials? If not, this is an important limitation as it is possible that differences in pre-trial diet have contributed to observed effects attribute to CAF. This should be acknowledged in the limitations section.
RESPONSE: Yes. Subjects were instructed to record and duplicate both the composition and timing of meals prior to each trial (e.g. meals not provided). While this is not a strict control, we believe that subject followed these instructions. This detail has been added to the manuscript and also mentioned in the limitations section.
The authors should consider providing the rationale for the metabolomics studies – this is not apparent. Also, it would be beneficial to outline why the 120 min sample was analysed as opposed to a fasting sample or other time points during the OGTT.
RESPONSE. We agree. A rationale has now been provided. Ideally, all samples would have been analyzed by metabolomics. However, this was not financially feasible. As such, we chose a time point that would likely show differences in caffeine and caffeine related metabolites. Since the half-life of caffeine is in the range of 4-6h, we chose the last time point of the OGTT, or 120min. To our surprise, the metabolomics data supported our genetics related findings nicely (e.g. OGTT differences) by showing distinction between slow and fast metabolizers in terms of circulating caffeine and caffeine related metabolites.
In addition to presenting the AUC data for the hormones in Table 2, it would be beneficial to show the time course data during the OGTT – similar for the glucose and insulin data in Figure 1.
RESPONSE: As the hormone responses were not significant, they were included in a table and not a figure in the manuscript. In case readers are interested, we have now included this information in a data supplement.
It is interesting that the C-peptide response (AUC data Table 2) was similar between the CAFF and DECAF trials. This indicates that insulin secretion was not affected by CAFF. Thus, despite the higher glucose levels during the OGTT in the CAFF trial, insulin secretion was not exacerbated. As one would expect the higher glucose response to cause greater insulin secretion, does this indicate that CAFF could somehow be impairing beta cell function? Have the authors considered using approaches such as the C-peptide minimal model to examine this? This point seems worthy of some discussion in the manuscript. The other point that these C-peptide results highlight is that the elevated insulin response caused by CAF must be mediated by effects on insulin clearance rather than insulin secretion. This point should also be explored. In this regard, it might be worth calculating indices of insulin clearance.
This is an excellent suggestion. I went back through the data and calculated two indices (Cersosimo et al. Assessment of Pancreatic & B-Cell Function: Review of Methods and Clinical Applications. Curr. Diabetes Rev. 10(1): 2–42. 2014). The first was the insulinogenic index of the oral glucose tolerance test. There was no statistically significant difference between the treatments (paired t-test, p= 0.28). Next, the Disposition Index, a useful method for determining B-cell function was examined. Again, no differences between treatments were found (paired t-test, p= 0.11). These findings fit with the overall consensus that they primary site of caffeine induced insulin resistance is a reduction in insulin-stimulated glucose uptake at skeletal muscle. This point has been emphasized in the revised manuscript.
Detail about the method for determining ISI is missing. Please include in the methods section.
RESPONSE: The appropriate reference for ISI has now been cited.
In relation to the limitations section, there are perhaps a couple of other points that could be considered. Firstly, venous blood was sampled, not arterialised venous blood from a heated hand vein. This could influence the glucose measurements. Secondly, an OGTT was used. While the OGTT has clinical relevance, its relevance to everyday life is somewhat limited since it is devoid of fat and protein, both of which can affect glycemic and hormone responses.
RESPONSE: We completely agree. These points have been added. Again, your comments on the manuscript were excellent and on point. Thank you!
Reviewer 2 Report
Introduction
Line 42: Reference number 6 does not appear to discuss impaired insulin sensitivity in adolescents specifically. I would opt to make this a separate sentence and explain impaired insulin sensitivity has been found in adults, thus it is worthy of exploring adolescents. This study by Smith et al. might also be useful: https://www.cambridge.org/core/journals/british-journal-of-nutrition/article/glucose-control-upon-waking-is-unaffected-by-hourly-sleep-fragmentation-during-the-night-but-is-impaired-by-morning-caffeinated-coffee/398A3EDA8C30EC89ADBB4C74C8E244B0 - they show caffeine after a disrupted sleep impairs glucose regulation in adults. The introduction states that disrupted sleep is a result of caffeine ingestion in adolescents, so this may create a vicious cycle – poor sleep = more caffeine. Since this results in worse glucose regulation in adults, caffeine’s effects on glucose metabolism is important to investigate in adolescents. This information could probably be included in the following paragraph (lines 45-60), which I again would recommend to make clear that the data are from adults, i.e. emphasise the very valid rationale of the study. (strong recommendation to make it clear that previous caffeine-glycaemia work was in adults)
Line 42-43: I would opt to stay focused on the actual rationale; since this study does not test the effects of withholding caffeine on other behaviours (smoking/social media use) I would opt to remove this sentence.
Line 52: “given this date” should read “given these data” as data is the plural of datum.
Materials and methods
Participants
Could you provide details of the power calculation, or what the sample size was based on? Since adolescents vary in maturation and glucose tolerance, n = 20 seems quite a small sample so it would be good to know how this was determined. (strong recommendation, or at least an acknowledgement that this was due to another limitation such as funding)
Test beverage
Where these taste matched drinks? E.g. sweetness can induce a cephalic phase insulin response and therefore alter outcomes. (strong recommendation, to at least acknowledge whether or not taste/other sensory experiences were matched)
Experimental design
States trials were separated by 1-4 weeks. I assume this is to account for the menstrual cycle where appropriate, could you confirm or elaborate in the manuscript why 1-4 weeks? If it was due to the menstrual cycle, could you please provide details (e.g. “all girls were tested in the estimate follicular phase 3-10 day after menses”)? (strong recommendation)
Statistical analysis
Line 141: Could you add a brief sentence regarding what assumptions were checked and how in order to determine the use of parametric or non-parametric analyses?
Results
Participants characteristics
I would suggest a table of participant characteristics with both overall sample, and then boys and girls separately. I feel this is important due to sex-differences in maturation, though it was really good to see sex-differences in outcomes were tested. I would also be interested in knowing how many girls were on hormonal contraceptives (or state in the methods explicitly that these were excluded). (I appreciate characteristics were reported in the methods as being available in another publication, but this makes reading the paper harder) (strong recommendation)
Impact of CED on glucose and insulin responses
Line 167: is this p-value for the interaction? Please clarify (strong recommendation)
Line 177: please provide the mean ± SD for ISI (strong recommendation – this is useful for e.g. meta-analyses)
Figure 1: The figures are quite small; recommend to stack them in 2 columns so they can be expanded a bit more for easier reading.
Figures 1B and 1D: suggest to amend Y axes to read “Glucose AUC (mmol/L*120min)” and “Insulin AUC (pmol/L*120min)”, respectively, for easier reading
Impacts of Caffeine Containing Energy Shots on Gut and Metabolic Hormone Secretion
Please clarify if the p-value is the anova interaction. I also think the p-value is supposed to read “p > 0.05”?
Combined caffeine sensitivity allele score
I would be interested to see the time trend (like Figure 1) for insulin and glucose between fast and slow metabolisers, though I appreciate this is vastly underpowered (maybe add to supplementary material?)
Discussion
Line 253-260: as above, it would be clearer to explicitly write whether the comparison studies are in adults, e.g. “these findings are in accordance with previous work in adults…” – I think this helps the reader to make a fairer assessment of the context as well as highlighting that you have actually done something quite novel. (strong recommendation)
Line 346: I fully agree that interactions in energy drink ingredients need to be considered especially considering the counteractive effects on some parameters (e.g. GABA activation with taurine versus inhibition with caffeine). The reference cited (46) states that there is variety of combinations that need to be considered, so it is unclear why the focus is on taurine-caffeine interactions (particularly, as the reference states, the levels of taurine are very low, probably not clinically meaningful). I would therefore opt to highlight in this sentence the range of interactions rather than focus just on taurine. I would also be keen to see some brief comparison (maybe in supplementary material?) of the drinks used to common energy drinks so we can more easily assess the validity of this experiment against drinks consumed in real life (e.g. highlighting whether the proportion of different ingredients was broadly similar to RedBull, Monster, etc (whatever is popular in Canada/USA)).
Line 355: suggest to remove “would contribute to the promotion of obesity” since this study tested glucose metabolism (not appetite or energy expenditure); plus if anything, insulin resistance causes weight loss due to malabsorption of glucose. A focus on glucose intolerance seems much more apt (strong recommendation)
Author Response
REVIEWER 2. We wish to sincerely thank the reviewer for their efforts in evaluating this manuscript, especially during the COVID pandemic that has increased stress levels worldwide. Your comments and attention to detail have increased the clarity and quality of our manuscript. All comments have been addressed and where possible changes to the manuscript have been made.
Line 42: Reference number 6 does not appear to discuss impaired insulin sensitivity in adolescents specifically. I would opt to make this a separate sentence and explain impaired insulin sensitivity has been found in adults, thus it is worthy of exploring adolescents.
This study by Smith et al. might also be useful: https://www.cambridge.org/core/journals/british-journal-of-nutrition/article/glucose-control-upon-waking-is-unaffected-by-hourly-sleep-fragmentation-during-the-night-but-is-impaired-by-morning-caffeinated-coffee/398A3EDA8C30EC89ADBB4C74C8E244B0 - they show caffeine after a disrupted sleep impairs glucose regulation in adults. The introduction states that disrupted sleep is a result of caffeine ingestion in adolescents, so this may create a vicious cycle – poor sleep = more caffeine. Since this results in worse glucose regulation in adults, caffeine’s effects on glucose metabolism is important to investigate in adolescents. This information could probably be included in the following paragraph (lines 45-60), which I again would recommend to make clear that the data are from adults, i.e. emphasise the very valid rationale of the study. (strong recommendation to make it clear that previous caffeine-glycaemia work was in adults)
RESPONSE: Thank you for pointing this out. Our first reference is indeed referencing University students (age 21-30y). To remedy this and make the references more specific to adolescents, two alternative references as follows have been added to the manuscript as follows:
Calamaro, C.J.; Mason, T.B.A.; Ratcliffe, S.J. Adolescents living the 24/7 lifestyle: Effects of caffeine and technology on sleep duration and daytime functioning. Pediatrics 2009, 123, doi:10.1542/peds.2008-3641.
Owens, J.A.; Mindell, J.; Baylor, A. Effect of energy drink and caffeinated beverage consumption on sleep, mood, and performance in children and adolescents. Nutr. Rev. 2014, 72, 65–71, doi:10.1111/nure.12150.
Line 42-43: I would opt to stay focused on the actual rationale; since this study does not test the effects of withholding caffeine on other behaviours (smoking/social media use) I would opt to remove this sentence.
RESPONSE: The citations regarding other behaviours was used to emphasize literature showing energy drinks are not health choices for children or adolescents. The sentence in question has now been removed.
Line 52: “given this date” should read “given these data” as data is the plural of datum.
RESPONSE: Correction has been made.
Materials and methods. Participants. Could you provide details of the power calculation, or what the sample size was based on? Since adolescents vary in maturation and glucose tolerance, n = 20 seems quite a small sample so it would be good to know how this was determined. (strong recommendation, or at least an acknowledgement that this was due to another limitation such as funding)
RESPONSE: The N per group was calculated using power calculations based on a power level of 0.8 and an alpha level of 0.05 with the difference between means and the standard deviations used in the power calculations derived from prior experiments (OGTT) from our laboratories using caffeine as a manipulation in adults. These indicated 10-12 participants were required. However, given the variability associated with adolescents (and their higher potential rate of drop-out), we opted to increase this to 20.
Test beverage. Where these taste matched drinks? E.g. sweetness can induce a cephalic phase insulin response and therefore alter outcomes. (strong recommendation, to at least acknowledge whether or not taste/other sensory experiences were matched)
RESPONSE: Yes, the beverages were very well matched in terms of volume, colour and taste. This has been highlighted in the revised manuscript.
Experimental design. States trials were separated by 1-4 weeks. I assume this is to account for the menstrual cycle where appropriate, could you confirm or elaborate in the manuscript why 1-4 weeks? If it was due to the menstrual cycle, could you please provide details (e.g. “all girls were tested in the estimate follicular phase 3-10 day after menses”)? (strong recommendation)
RESPONSE: Trial order was randomly assigned by a computer-generated randomizing program. However, we did not control for menstrual cycle phase as previous work has shown that ‘menstrual cycle, gender, exercise, and thermal stress have no effect on the absorption, distribution, metabolism, and elimination of caffeine when reproductive status and environmental and dietary factors were controlled’ (McLean et al, J. Appl. Physiol, 2002). This information and reference has been added to the revised manuscript. Instead, we chose 1 week on the lower end to ensure washout and 4 weeks as an upper range to limit variation. In truth, a longer study duration was also to accommodate parents who drove and often accompanied participants trials.
Statistical analysis. Line 141: Could you add a brief sentence regarding what assumptions were checked and how in order to determine the use of parametric or non-parametric analyses?
RESPONSE: These details have now been added to the manuscript.
Results. Participants characteristics. I would suggest a table of participant characteristics with both overall sample, and then boys and girls separately. I feel this is important due to sex-differences in maturation, though it was really good to see sex-differences in outcomes were tested. I would also be interested in knowing how many girls were on hormonal contraceptives (or state in the methods explicitly that these were excluded). (I appreciate characteristics were reported in the methods as being available in another publication, but this makes reading the paper harder) (strong recommendation)
RESPONSE: Being on oral contraceptives was an exclusion of participation. This has now been added to the manuscript. As mentioned above, ‘menstrual cycle, gender, exercise, and thermal stress have no effect on the absorption, distribution, metabolism, and elimination of caffeine when reproductive status and environmental and dietary factors were controlled’ (McLean et al, J. Appl. Physiol, 2002). Additional information regarding the subjects has also been added as a data supplement as recommended.
Impact of CED on glucose and insulin responses. Line 167: is this p-value for the interaction? Please clarify (strong recommendation). Line 177: please provide the mean ± SD for ISI (strong recommendation – this is useful for e.g. meta-analyses)
RESPONSE: Thank you for this suggestion. Values have been added to the manuscript.
Figure 1: The figures are quite small; recommend to stack them in 2 columns so they can be expanded a bit more for easier reading.
RESPONSE: Agreed. Figures have now been changed to a 2 by 3 stack to increase both size and clarity.
Figures 1B and 1D: suggest to amend Y axes to read “Glucose AUC (mmol/L*120min)” and “Insulin AUC (pmol/L*120min)”, respectively, for easier reading
RESPONSE: Wonderful suggestion, these changes have been made.
Impacts of Caffeine Containing Energy Shots on Gut and Metabolic Hormone Secretion. Please clarify if the p-value is the anova interaction. I also think the p-value is supposed to read “p > 0.05”?
RESPONSE: This is indeed supposed to be p >0.05 and has been corrected.
Combined caffeine sensitivity allele score. I would be interested to see the time trend (like Figure 1) for insulin and glucose between fast and slow metabolisers, though I appreciate this is vastly underpowered (maybe add to supplementary material?)
RESPONSE: Agree, this is an interesting aspect. I went back through the data looking for time trends and came to the conclusion that there is simply not enough resolution in the data. There maybe differences, but they are not apparent within the scope of the OGTT. I suspect more time points are needed to distinguish. As is, the only difference between genotypes is in the magnitude and slope of the curves.
Discussion. Line 253-260: as above, it would be clearer to explicitly write whether the comparison studies are in adults, e.g. “these findings are in accordance with previous work in adults…” – I think this helps the reader to make a fairer assessment of the context as well as highlighting that you have actually done something quite novel. (strong recommendation)
RESPONSE: Thank you. We agree, to increase clarity the above-mentioned section has been clarified to read ‘in adults’.
Line 346: I fully agree that interactions in energy drink ingredients need to be considered especially considering the counteractive effects on some parameters (e.g. GABA activation with taurine versus inhibition with caffeine). The reference cited (46) states that there is variety of combinations that need to be considered, so it is unclear why the focus is on taurine-caffeine interactions (particularly, as the reference states, the levels of taurine are very low, probably not clinically meaningful). I would therefore opt to highlight in this sentence the range of interactions rather than focus just on taurine. I would also be keen to see some brief comparison (maybe in supplementary material?) of the drinks used to common energy drinks so we can more easily assess the validity of this experiment against drinks consumed in real life (e.g. highlighting whether the proportion of different ingredients was broadly similar to RedBull, Monster, etc (whatever is popular in Canada/USA)).
RESPONSE: While we agree that drink comparison is of interest, we do not have the time or the resources to pull this information in the required timeframe (revisions Due Dec 6 - today). While taurine in energy drinks is below concentrations expected to elicit adverse reactions, interactions with caffeine have been found. This is highlighted in the existing reference. To further support taurine interactions, we have also added an additional reference (Steinke et al, Annals of Pharmacotherapy, 2009).
Line 355: suggest to remove “would contribute to the promotion of obesity” since this study tested glucose metabolism (not appetite or energy expenditure); plus if anything, insulin resistance causes weight loss due to malabsorption of glucose. A focus on glucose intolerance seems much more apt (strong recommendation)
RESPONSE: The sentence in question has been removed from the revised manuscript.